# Phosphodiester Stationary Phases as Universal Chromatographic Materials for Separation in RP LC, HILIC, and Pure Aqueous Mobile Phase

**DOI:** 10.3390/ma16093539

**Published:** 2023-05-05

**Authors:** Mikołaj Dembek, Szymon Bocian

**Affiliations:** Chair of Environmental Chemistry and Bioanalysis, Faculty of Chemistry, Nicolaus Copernicus University, 7 Gagarina Str., 87-100 Torun, Poland

**Keywords:** polar-embedded stationary phases, pure water, nucleosides, liquid chromatography

## Abstract

Modern analytical chemistry techniques meet the need for greater attention to ecological and economic aspects. It is becoming necessary to seek solutions to reduce harmful waste production, especially in large quantities. High-performance liquid chromatography is a technique widely used in many industries, including mainly pharmaceuticals, and requires an approach to reduce the significant amount of organic solvent waste. One of the green chemistry solutions is using environmentally benign substitutes, such as pure water, supercritical dioxide, and ethanol. Our work focuses on the preparation and application of new stationary phases with embedded hydrophilic groups for separations using pure water in liquid chromatography. Polar-embedded stationary phases are obtained by attaching a phosphodiester group and 4 different hydrophobic molecules. The studies consisted of hydrophobicity measurements, concentration dependence of retention of the organic additive to the mobile phase, and chromatographic separations of polar and non-polar substance mixtures in RP-LC and HILIC systems. Three mixtures were studied: purine alkaloids, benzene, and polycyclic aromatic hydrocarbons and nucleosides. The stationary phases interact differently with the analytes depending on the attached hydrophobic group. It is possible to use pure water to separate each mixture under study. It is also significant that it has been possible to separate a mixture of completely non-polar compounds using pure water for the first time. The research being carried out is crucial in synthesizing new polar-embedded stationary phases, providing work versatility and high environmental performance.

## 1. Introduction

Modern high-performance liquid chromatography is based on two retention modes that differ in the mechanism—reversed-phase system (RP) and hydrophilic interaction chromatography (HILIC). Nowadays, the normal-phase system (NP) is rarely used due to a large amount of harmful organic solvents [1]. The essence of chromatographic separation is provided by interactions occurring at the surface of the stationary phase between the analyte and the solvent, the analyte and the stationary phase, and the solvent and the stationary phase [2,3,4].

A reversed-phase system (RP LC) is used to analyze non-polar compounds. A non-polar stationary phase in the form of silica, polymers, or modified silica with alkyl chains (e.g., C8, C18) is rinsed with a polar mobile phase usually consisting of water and methanol or acetonitrile. The analytes are separated in terms of their hydrophobicity. An adsorption/partition mechanism predominates here, in which the non-polar parts of the analytes are attracted to the hydrophobic ligands of the stationary phase, which induces retention. Elution of the analytes follows their decreasing polarity. Polar stationary phases and a mobile phase with a high content of organic modifiers and buffers are used in HILIC. This reduces hydrophobic interactions between the sample and the stationary phase, allowing more polar substances to be separated and eluted. The mechanism is based on partitioning the analyte between adsorbed water layer on the stationary and mobile phases. In the normal phase system, the stationary phase is polar—neat silica—while the mobile phase is non-polar—hexane, heptane, cyclohexane. The mechanism is based on the adsorption of polar analytes on the silica surface, where silanol and siloxane groups are located. The more polar the analyte, the stronger the interaction; thus, the elution of a particular compound will be later. When using bonded stationary phases, the primary mechanism is partitioning the analyte between the bulk of the mobile phase and the groups chemically bonded to the silica surface [5,6,7].

When the stationary phase has bonded hydrophobic groups, it is possible to retain non-polar compounds whose interactions with the organic-water mobile phase are weaker than those with non-polar groups. We are then dealing with an inverted phase system (RP). Increasing organic water content increases retention because non-polar substances have a higher affinity for bound hydrophobic groups. The mechanism can either partition when the analyte gets between the alkyl chains attached to the silica or adsorption when they adsorb onto the alkyl-silica surface [5]. Analysis of polar compounds requires modifying silica with polar groups. We then deal with hydrophilic interaction chromatography (HILIC) [8,9,10]. Using a high content of an organic modifier in the mobile phase causes a small water content to form a so-called “hydrophilic pillow” at the silica surface. It allows analyte partitioning between the bulk of the mobile phase and the water adsorbed near the surface. Increasing the water content of the mobile phase decreases retention as the mobile phase becomes more polar, so the affinity of the hydrophilic analyte increases [6,7,11,12].

The separation of polar and non-polar compounds requires both different stationary phases and different mobile phase conditions. It raises the question: is it possible to prepare such a stationary phase that allows simultaneous operation in RP and HILIC? Phases with embedded polar groups, in which hydrophilic groups and hydrophobic parts in the form of alkyl chains, aromatic rings, or whole weakly polar molecules are simultaneously present, are of increasing interest. There are several ways to attach polar groups. They can be bonded at the stage of secondary silanization (polar-end-capped), embedded between the silica surface and the hydrophobic group (polar-embedded), or attached to the end of the non-polar group (polar-headed) [13,14,15,16]. They exhibit a mixed retention mechanism and can work at high organic modifier content separating polar compounds in HILIC and at high water content separating non-polar compounds in RP [17,18,19,20,21,22].

Modern industries using liquid chromatography produce significant amounts of harmful organic solvent waste. The unit waste generation by a chromatograph is insignificant; however, multiplying this by the number of instruments working in laboratories and the amount of time they operate, the values increase significantly [23]. Therefore, looking for solutions to reduce harmful and sometimes even toxic residues is crucial. The basis is to follow the principle of green chemistry—reduce, replace, recycle. So, the solution may be to replace harmful solvents with greener alternatives. Among them are ethanol, supercritical carbon dioxide, and water. In order to use such replacements, it is necessary to introduce a suitable stationary phase to work under these conditions [1,13,24]. Thus, there is a requirement in liquid chromatography for the synthesis and broad characterization of new materials that allow work under conditions that provide ecological benefits. There are many reports in the Literature regarding stationary phases operating in mixed retention mode [25,26,27,28]. Recent solutions that provide such operation are polar-embedded and polar-end-capped stationary phases. They have both polar and non-polar parts attached. The synthesis method as well as the appropriate selection of attached groups provides the possibility to moderate the selectivity as well as the working range of the prepared material. This opens up possibilities for the preparation of an incredible number of new stationary phases with diverse properties and applications. It is also possible to obtain such materials that will provide efficient and selective work in pure water conditions.

In our work, we consider pure water the only mobile phase component. New stationary phases with embedded polar groups provide the possibility of such analyses. By appropriate selection of the polar group and the attached hydrophobic group, selectivity can be controlled [17,18,20,29,30]. Thus, four stationary phases have been prepared in which a phosphodiester group fulfills the role of the hydrophilic group. In contrast, the non-polar group consists of four different substituents: an octadecyl chain, a decyl chain, an aromatic ring, and a cholesterol molecule. This work focuses on the presentation of new stationary phases and a broad description of their chromatographic properties. The prepared materials were characterized in terms of operation in both RP and HILIC modes, which allowed the separation of polar and non-polar groups of compounds, as well as effective work in “green chromatography” conditions, that is, in pure water.

## 2. Materials and Methods

### 2.1. Equipment and Chemicals

All chromatographic analyses were performed using the Shimadzu Nexera UHPLC system (Kioto, Japan). This chromatograph is equipped with a binary solvent delivery system (LC-30AD), an autosampler with a 20 µL volume loop (SIL-20AC), a column thermostat (CTO20AC), and a diode-array UC-detector (SPD-M20A). LabSolution LC/GC 5.65 software (Shimadzu, Kioto, Japan) was used to collect and process data and control the apparatus. Acetonitrile (ACN) (HPLC Grade) and methanol (MeOH) (HPLC Grade) were purchased from Sigma–Aldrich (Steinheim, Germany). Packaging solvents, isopropanol, and methanol were purchased from J.T. Baker, Deventer, the Netherlands. Water was prepared with a Milli-Q Water Purification System (Millipore Corporation, Bedford, MA, USA). Standards of nucleosides—adenosine, guanosine, uridine—were obtained from Applichem (Darmstadt, Germany). Non-polar compounds—naphthalene, benzene, phenanthrene, caffeine, theophylline, and theobromine—were obtained from Sigma–Aldrich (St. Louis, MO, USA). For the synthesis of stationary phases (3-glycidoxypropyl) trimethoxysilane, decanol, octadecanol, cholesterol, benzyl alcohol, and phosphorus chloride were used, which were purchased from Alfa Aesar (Karlsruhe, Germany). Haskel (Burbank, CA, USA) laboratory equipment and packing pump were used to pack the stationary phases into 125 × 4.6 mm long empty columns.

### 2.2. Materials

Four stationary phases with embedded phosphodiester groups were synthesized for the study. They were prepared on Kromasil 100 Å silica (Akzo Nobel, Bohus, Sweden), with a diameter of 5 µm, a specific surface area of 320 m^2^/g, an average pore size of 11 nm, and a pore volume of 0.9 cm^3^/g. The phases were prepared by attaching a short linker in the form of a propyl chain and then a phosphodiester group to the surface silanols. Each of the stationary phases has a different hydrophobic molecule attached to a phosphodiester group: Diol-P-C10, a decyl chain; Diol-P-C18, an octadecyl chain; Diol-P-benzyl, a benzyl group; and Diol-P-chol, a cholesterol molecule. A detailed procedure for synthesizing and characterizing these materials has been described in previous works [31,32], while the structures are shown in Figure 1.

### 2.3. Methods

The stationary phases were packed using the slurry method. 1.7 g of each phase was suspended in 15 mL of isopropanol, then shaken for 1 min and placed in an ultrasonic bath for 10 min before packing itself. Packing was performed at 400 bar for 2 h, and methanol was used as the pressing solvent. After this time, the pressure was reduced to 100 bar, and the column was flushed with 90/10, 50/50, and 10/90 ACN/H_2_O solutions for 1 h each. The final packed column was left in pure methanol. The packaging optimization was described in a previously published paper.

The hydrophobicity of stationary phases was performed based on a modified Galushko test [33]. For this purpose, toluene and benzene samples prepared in a 40:60 MeOH:H_2_O solution were used. Instead of a mobile phase of 60:40 MeOH:H_2_O composition, the same solvents of 40:60 MeOH:H_2_O composition were used. This was due to the low retention of analytes. Hydrophobicity (H_G_) was calculated based on Equation (1). The dead volume was measured for an excess amount of one of the mobile phase components: acetonitrile.
H_G_ = (k_toluene + k_benzene)/2,(1)
where H_G_ is hydrophobicity, k_toluene is the retention factor of toluene, and k_benzene is the retention factor of benzene. The dependence of retention on the mobile phase composition was determined for a polar compound: guanosine, and a non-polar compound: naphthalene. Each sample was analyzed at increasing concentrations (% *v*/*v*) of acetonitrile in water (0, 2, 5, 10, 15, 25, 35, 45, 55, 65, 75, 85, 90, 95, 98, 100) by injecting 0.1 µL of guanosine solution and 1 µL of naphthalene solution. The mobile phase was prepared by an on-line mixer. Analyses were performed at 30 °C. The signal was detected with a UV-Vis detector at 254 nm.

Single compound analyses were performed to determine retention and select mixtures’ substances. The analysis conditions were the same as those for determining the dependence of retention on the mobile phase composition. Analyses were performed for nucleosides, purine alkaloids (caffeine and its derivatives), benzene, and polycyclic aromatic hydrocarbons. All samples were prepared in a 50/50 (*v*/*v*) acetonitrile-water solution. Dead time was determined for acetonitrile at a mobile phase composition of 60/40 ACN/H_2_O. For each mixture, analyses were carried out in the RP LC system—100% water and HILIC—acetonitrile concentration selected for best separation.

All chromatographic analyses, both for hydrophobicity measurements, retention-mobile phase concentration relationships, as well as separation analyses, were performed at least three times. In the results, it did not make sense to include error bars on the graphs, as they were not visible at this scale.

## 3. Results and Discussion

### 3.1. Hydrophobicity

Novel chromatographic materials have to be characterized. In order to describe the surface properties of the stationary phases used, a Galushko test [33] was performed to determine the hydrophobicity and silanol activity. The first property is presented in Table 1. The silanol activity could not be determined due to ion-exchange interactions of aniline. On the other hand, determining silanols on such polar embedded stationary phase provides information about all polar groups, e.g., diol, not only silanols. The phase with an attached benzyl group proved to be the least hydrophobic. It may be due to a delocalized electron cloud in the aromatic ring and a small number of carbon atoms compared to the other phases. The phase with an attached octadecyl chain proved to be the most hydrophobic. Less polar than Diol-P-C18 is the Diol-P-chol phase, most likely due to its size and steric efficiency it shields less well the polar effect of the phosphodiester group.

### 3.2. Retention Analyses

In the next stage of the study, analyses were conducted to determine retention’s dependence on the acetonitrile concentration in the acetonitrile-water mobile phase. For this purpose, naphthalene was chosen as a non-polar compound and guanosine as a polar compound. The retention trends are shown in Figure 2. The non-polar compound showed increased retention with increasing water content in the mobile phase (Figure 2A). This typical mechanism occurs in RP-LC. A difference in retention can be seen depending on the stationary phase used. The attachment of the benzyl group exhibited the lowest retention, while the cholesterol molecule showed the highest retention. The appearance of naphthalene retention in pure water for each of the studied stationary phases confirms the mixed retention mechanism. Both hydrophilic and hydrophobic interactions as well as secondary interactions are involved in retention. The varying retention time on each phase is due to their differences in hydrophobicity. An increase in retention is observed with an increase in hydrophobicity.

Guanosine retention occurs at both ends of the percentage of water content in the mobile phase. Under highly aqueous conditions, the retention is small compared to the high acetonitrile concentration. It is important to note that all tested phases allow retention in RP LC and HILIC. The differences between the retention in each phase are due to their structure and the interactions involved in the mechanism. It is also evident that on the RP LC side, the order of the phases with increasing retention differs from the arrangement when analyzing naphthalene. This is because other types of interactions contribute more to the retention mechanism. In the case of high water content, a so-called “hydrophilic pillow” is formed, which allows polar compounds to be retained near the surface of the mobile phase [6]. The presence of phosphodiester groups at the surface of each phase promotes the formation of this “pillow” and its attainment of a greater height than if it were formed only with the participation of free silanols presented on the silica surface. The high acetonitrile content provides retention according to the well-known HILIC mechanism. The low hydrophobicity of the Diol-P-benzyl column indicates the high amount of free silanols and unfunctionalized diols. This makes hydrophilic interactions the strongest, and retention of guanosine is the highest on Diol-P-benzyl among the columns tested. However, it has to be emphasized that other analytes in HILIC, at different mobile phase compositions, may create other retention orders.

### 3.3. Mixture Separation

To confirm the ability to separate polar and non-polar compounds in pure water and to separate polar mixtures in HILIC mode, the following mixtures were prepared: (A) caffeine, theophylline, and theobromine; (B) adenosine, guanosine, and uridine; (C) benzene, naphthalene, and phenanthrene.

#### 3.3.1. Purine Alkaloids

The results of the mixture (A) separation in pure water are shown in Figure 3.

Applying pure water as a mobile phase allows the complete elimination of organic solvents, making it the "greenest” solution in chromatography. Despite using a single-component mobile phase, it is possible to separate the mixture’s components in each tested stationary phase. The separation to the baseline was possible on the Diol-P-C10 and Diol-P-benzyl columns. They also had the shortest retention times, making separation possible in less than 4 min. It is due to their lower hydrophobicity. At the same time, Diol-P-C18 and Diol-P-chol phases provided longer retention times. There was also no separation of theophylline from theobromine to the baseline. It confirms the effect of the hydrophobic group on separation selectivity.

Using the same columns and the purine alkaloid mixture, separating them in HILIC mode was possible. The optimal concentration of ACN/H_2_O was selected for each column. Chromatograms are shown in Figure 4.

Only the Diol-P-benzyl column allowed the separation of the mixture to the baseline. It is the stationary phase with the lowest hydrophobicity. The differences in the structure of the caffeine derivatives include the different amounts and locations of the methyl group in the imidazolopyrimidine ring. Using phases with C10 and C18 chains attached did not allow the separation of either component to the baseline. In these two cases, it was necessary to use a higher water content than in the case of the Diol-P-benzyl and Diol-P-chol phases. This was due to the fact that the retention of the mixture’s components increased differently with a change in ACN concentration. At lower concentrations, the first two components eluted together, while at higher concentrations, the second and third components eluted together, making separation impossible. Using a buffer to stabilize the pH and ionic strength would likely allow narrower peaks and separation to the baseline in each case.

#### 3.3.2. Nucleosides

The second group of tested polar compounds was nucleosides. The mixture included uridine, guanosine, and adenosine. Each of the columns analyzed allowed separation of the mixture components using pure water as the only mobile phase component. The Diol-P-benzyl column performed the worst, as it failed to separate uridine and guanosine to the baseline. The Diol-P-chol column yielded more than 14,000 theoretical plates for the least retained component of the mixture (uridine), which is comparable to commercial columns operating in pure water. The chromatograms for this phase are shown in Figure 5. The elution order of the compounds in each case was the same when using pure water and in HILIC. This is untypical behavior because the retention order in HILIC and RP conditions is usually the opposite. This may indicate the strong influence of the phosphodiester group, present in each phase, on elution.

Applying a high acetonitrile content mobile phase also had the desired effect of separating the mixture. The worst phase was Diol-P-C10, where elution times were very long, and the tailing of peaks affected lengthy analysis times reaching up to 60 min. The Diol-P-C18 phase yielded a reduced plate height (h) within 3, which is considered outstanding efficiency [34]. The Diol-P-benzyl column in this phase system proved to perform much better. It succeeded in separating the mixture in less than 5 min using it. All results for the nucleoside analyses are summarized in Table 2.

#### 3.3.3. Benzene and Polycyclic Aromatic Hydrocarbons

In RP LC, the eluotropic array of solvents for separating non-polar compounds is reversed, so organic solvents such as acetonitrile and methanol have greater elution strength than water. Thus, at high organic solvent content, separation is not possible, as all substances will elute at a void time. Only by increasing the water content can retention be increased, and with proper column selectivity, a given mixture of compounds can be separated. In classical RP LC, a small presence of an organic modifier is necessary to enable elution, as using pure water to separate non-polar compounds would result in a lack of elution and the permanent retention of analytes in the silica bed. However, the use of polar-embedded stationary phases enables a mixed retention mechanism. As a result, it was possible to separate a mixture of completely non-polar compounds in pure water for the first time. Of course, the separation efficiency, retention time, or peak symmetry are unsatisfactory and compare inferiorly with the separation in RP LC by adding an organic modifier. However, the sole fact of using pure water without organic additives from an ecological and economic point of view is significant for modern liquid chromatography. An example chromatogram for the Diol-P-C10 column is shown in Figure 6, while all analysis parameters are summarized in Table 3.

All separations have been made to the baseline. Due to the very weak interactions of the analytes with the mobile phase, there was significant peak tailing in each case. On the less hydrophobic phases—Diol-P-C10 and Diol-P-benzyl—the separation was completed in about 10 min; however, using the more hydrophobic phases—with an octadecyl chain or cholesterol molecule attached—significantly increases the analysis time.

Analyzing all the results of the chromatographic analyses, it is apparent that stationary phases with embedded phosphodiester groups can work effectively to separate both polar and non-polar compounds in water. In addition, these phases operate on both sides of the organic modifier content range in combination with water in the mobile phase (RP LC and HILIC). This paper presents the possibilities of separating simple mixtures of polar and non-polar compounds. Preliminary studies for this work have shown that more complex mixtures of compounds of closer polarity may not be completely separated. In such a case, in order not to give up purely aqueous conditions, it is possible to use a temperature gradient already published many times [35,36,37]. The possibility of improving the results by, for example, stabilizing the pH or achieving a better-packed bed in the column arises. These phases offer great potential for modern chromatography, bearing in mind the ecology and economics of this branch of science and analytics. Research is currently underway to analyze β-blockers under both isocratic and gradient conditions. There is a plan to publish the obtained results in the near future.

## 4. Conclusions

This work focused on the preparation and description of chromatographic properties of stationary phases of the polar-embedded type, where the polar part was a phosphodiester group. An important part of the research was the chromatographic analysis allowing the separation of compounds of different polarities. The possibility of using each studied phase to separate a group of caffeine derivatives, nucleosides, benzene, and polycyclic aromatic hydrocarbons in pure water was confirmed. These results are groundbreaking, as they are the first stationary phases working in pure water with the ability to separate polar and non-polar compounds. In addition, their simultaneous ability to operate in a HILIC system with high efficiency makes these materials an incredible potential for today’s high-performance liquid chromatography. The use of such materials will significantly improve the current unfavorable environmental aspect of HPLC associated with the production of large quantities of organic solvents while at the same time opening the way for the use of a single material for multiple chromatographic applications. The considerable number of combinations of different polar groups with attached non-polar groups represents the immense opportunities posed by polar-embedded stationary phases. The results require further research to improve efficiency, selectivity, and peak symmetry.

## Figures and Tables

**Figure 1 materials-16-03539-f001:**
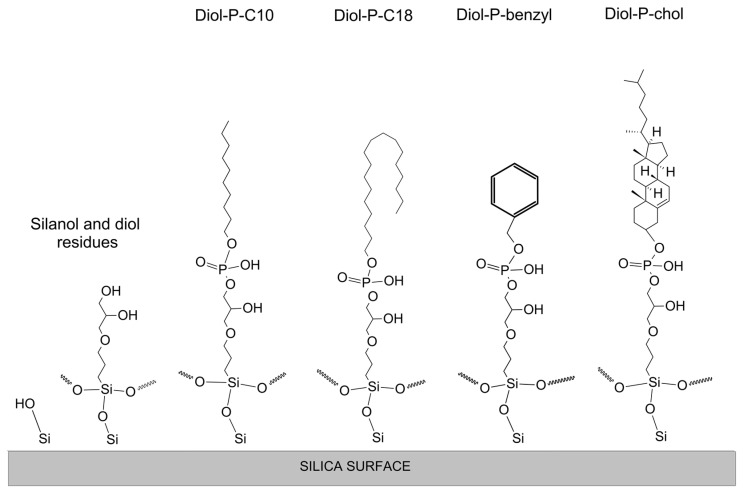
Schematic representation of four different stationary phases with embedded phosphodiester groups. On the left side of the figure, residual silanol and diol structures are shown as possible functional groups occurring in every phase.

**Figure 2 materials-16-03539-f002:**
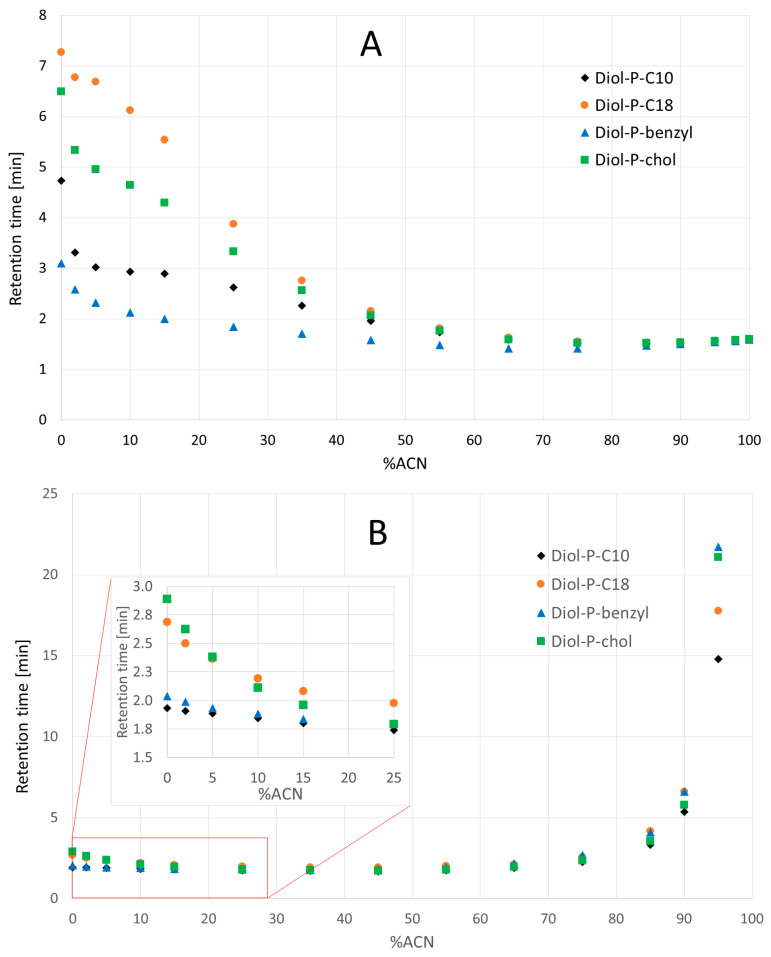
Dependence of naphthalene (**A**) and guanosine (**B**) retention time on the percentage of acetonitrile in the ACN/H2O mobile phase for stationary phases with incorporated phosphodiester groups.

**Figure 3 materials-16-03539-f003:**
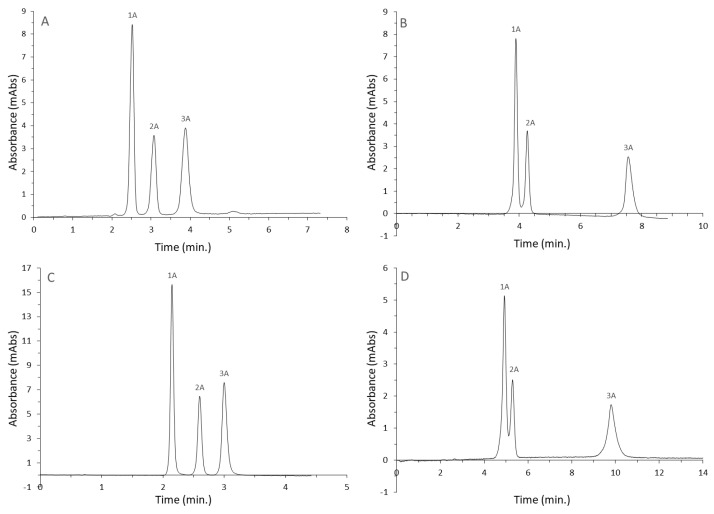
Separation of purine alkaloid in pure water on a series of chromatography columns: (**A**)—Diol-P-C10, (**B**)—Diol-P-C18, (**C**)—Diol-P-benzyl, (**D**)—Diol-P-chol, compounds: 1A—theophylline, 2A—theobromine, 3A—caffeine.

**Figure 4 materials-16-03539-f004:**
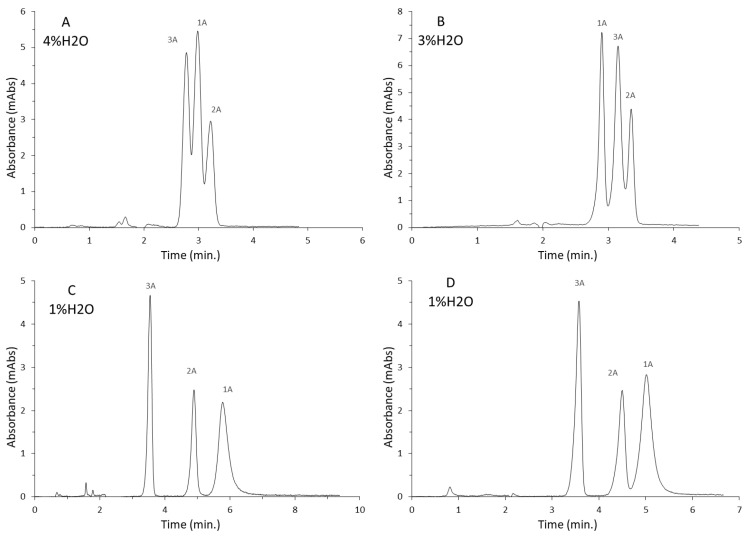
Separation of purine alkaloid in HILIC on a series of chromatography columns: (**A**)—Diol-P-C10, (**B**)—Diol-P-C18, (**C**)—Diol-P-benzyl, (**D**)—Diol-P-chol, compounds: 1A—theophylline, 2A—theobromine, 3A—caffeine.

**Figure 5 materials-16-03539-f005:**
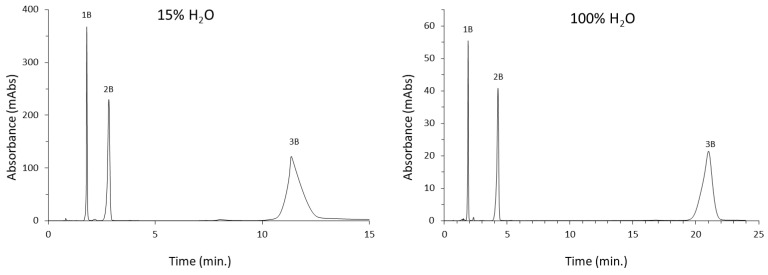
The separation of nucleosides in pure water and 15% of water on a Diol-P-C18 column. 1B—uridine, 2B—guanosine, 3B—adenosine.

**Figure 6 materials-16-03539-f006:**
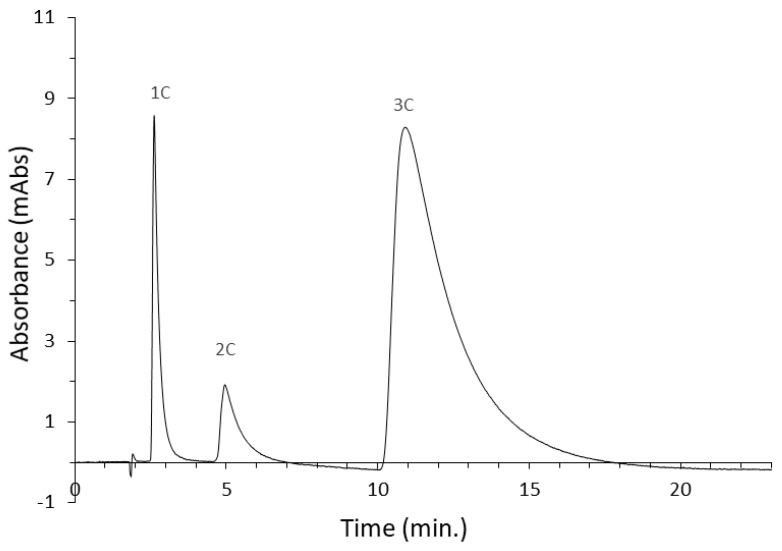
The separation of benzene and polycyclic aromatic hydrocarbons in pure water on a Diol-P-C10 column. 1C—benzene, 2C—naphthalene, 3C—phenanthrene.

**Table 1 materials-16-03539-t001:** Characteristics of bonded polar-embedded stationary phases. Hydrophobicity was determined based on the Galushko test.

Stationary Phase	Carbon Load [%]	Coverage Density [µmol/m^2^]	Hydrophobicity (Hg)
Diol-P-C10	3.43	0.56	0.149
Diol-P-C18	4.18	0.42	0.303
Diol-P-Benzyl	2.86	0.56	0.044
Diol-P-Chol	9.31	0.87	0.221

**Table 2 materials-16-03539-t002:** Summary of the results of separation of nucleosides in pure water and in HILIC on a series of columns with embedded phosphodiester groups. 1B—uridine, 2B—guanosine, 3B—adenosine.

Stationary Phase	Mobile Phase	Compound	R_t_	k	NTP	R_s_	As^0.1^
Diol-P-C10	100% H_2_O	1B	1.720	0.025	2737	9.383	0.981
		2B	1.907	0.136	2467	1.311	0.939
		3B	2.655	0.582	2324	4.004	0.916
	4% H_2_O	1B	5.774	2.441	62	-	2.141
		2B	14.079	7.390	102	1.952	3.022
		3B	25.476	14.182	255	1.904	2.155
Diol-P-C18	100% H_2_O	1B	1.782	0.087	9210	-	0.944
		2B	2.722	0.661	2667	6.591	1.061
		3B	20.373	11.430	1586	15.641	1.845
	15% H_2_O	1B	2.126	0.297	8197	-	0.879
		2B	4.357	1.658	8099	15.511	0.838
		3B	10.951	5.681	5292	16.572	1.397
Diol-P-benzyl	100% H_2_O	1B	2.727	0.606	659	2.668	2.36
		2B	4.949	1.914	366	3.042	5.398
		3B	8.487	3.998	102	1.612	7.274
	15% H_2_O	1B	2.466	0.452	5638	-	1.203
		2B	3.427	1.018	5064	5.933	1.249
		3B	4.204	1.476	3306	3.204	1.232
Diol-P-Chol	100% H_2_O	1B	1.800	0.068	14,503	15.136	0.921
		2B	2.828	0.679	3804	8.458	0.810
		3B	11.355	5.739	1403	12.216	1.478
	25% H_2_O	1B	1.908	0.133	6656	-	0.821
		2B	4.283	1.542	5240	14.382	0.730
		3B	21.023	11.477	2824	18.403	0.807

R_t_—retention time, k—retention factor, R_s_—resolution, NTP—number of theoretical plates, As^0.1^—asymmetry factor.

**Table 3 materials-16-03539-t003:** Summary of the results of benzene and polycyclic aromatic hydrocarbons separation in pure water on a series of columns with embedded phosphodiester groups. 1C—benzene, 2C—naphthalene, 3C—phenanthrene.

Stationary Phase	Compound	R_t_	k	NTP	R_s_	As^0.1^
Diol-P-C10	1C	2.610	0.555	1168	2.949	2.624
	2C	4.942	1.945	434	3.718	3.440
	3C	10.906	5.499	174	2.799	3.687
Diol-P-C18	1C	2.632	0.606	1842	4.083	4.082
	2C	6.519	2.978	714	6.367	4.894
	3C	33.867	19.663	1261	11.417	3.524
Diol-P-Benzyl	1C	2.727	0.606	659	2.668	2.360
	2C	4.949	1.914	366	3.042	5.398
	3C	8.487	3.998	102	1.612	7.274
Diol-P-Chol	1C	3.088	0.832	1447	4.706	1.943
	2C	7.111	3.220	1725	7.971	1.941
	3C	42.533	24.242	998	11.671	2.908

R_t_—retention time, k—retention factor, R_s_—resolution, NTP—number of theoretical plates, As^0.1^—asymmetry factor.

## Data Availability

The data presented in this study are available on request from the corresponding author.

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
