# Peer review of "Phosphodiester Stationary Phases as Universal Chromatographic Materials for Separation in RP LC, HILIC, and Pure Aqueous Mobile Phase"

_materials, 2023, doi:10.3390/ma16093539_

Round 1
Reviewer 1 Report
The authors presented in the manuscript the results on the preparation and evaluation of phosphodiester based stationary phases which can be used in both RPLC and HILIC modes. The concept of operating a single column in both RPLC and HILIC modes is not new. The authors should include some discussions on the published work on this concept in the introduction section. Although the phosphodiester based phases are newly prepared, more characterization data is needed before the manuscript can be accepted for publication. More specific comments are below:
1. Lines 138-139, the authors state that the silanol activities could not be determined. However, the surface coverage of the phosphodiester phases is relatively low. This seems to be inconsistent. Please explain.
2. The hydrophobicity of the prepared phases was characterized. However, the adsorbed water layer on the surface of the stationary phase, which is very important to HILIC separation, was not evaluated at all. Please provide data on the adsorbed water layer.
3. Line 181, the authors stated that the Diol-P-Benzyl column has the strongest retention due to its lowest hydrophobicity. However, the data in Table 2 shows that the retention factors of 2B and 3B on the Diol-P-C18 are higher than those on the Diol-P-benzyl phase. The Diol-P-Col phase was evaluated using 25% water, but the retention factors were also higher.
4. Lines 174-176, the description of the HILIC retention mechanisms is not accurate.
5. The discussion of general retention mechanisms in chromatography in the first paragraphs of the introduction section is vague and in some cases not accurate. It needs to be revised.
Author Response
The authors presented in the manuscript the results on the preparation and evaluation of phosphodiester based stationary phases which can be used in both RPLC and HILIC modes. The concept of operating a single column in both RPLC and HILIC modes is not new. The authors should include some discussions on the published work on this concept in the introduction section. Although the phosphodiester based phases are newly prepared, more characterization data is needed before the manuscript can be accepted for publication. More specific comments are below:
- Introduction was corrected.
- Lines 138-139, the authors state that the silanol activities could not be determined. However, the surface coverage of the phosphodiester phases is relatively low. This seems to be inconsistent. Please explain.
- Measurement is impossible due to ion-exchange interaction of protonated aniline with ionized phosphate. The ion-exchange properties of these phases is a topic of further work which be submitted to Molecules soon.
- The hydrophobicity of the prepared phases was characterized. However, the adsorbed water layer on the surface of the stationary phase, which is very important to HILIC separation, was not evaluated at all. Please provide data on the adsorbed water layer.
- We are working on the adsorption evaluation of tested stationary phases. We have water adsorption data, but we would like not to present it in this study. We can provide it as a material for Reviewer, not for publication.
- Line 181, the authors stated that the Diol-P-Benzyl column has the strongest retention due to its lowest hydrophobicity. However, the data in Table 2 shows that the retention factors of 2B and 3B on the Diol-P-C18 are higher than those on the Diol-P-benzyl phase. The Diol-P-Col phase was evaluated using 25% water, but the retention factors were also higher.
- We agreed with the Rewiever. It was about guanosine. More detailed conclusion is provided.
- Lines 174-176, the description of the HILIC retention mechanisms is not accurate.
- Hydrophilic pillow described there is in pure water. It look like HILIC, but we are on the other site of mobile phase composition.
- The discussion of general retention mechanisms in chromatography in the first paragraphs of the introduction section is vague and in some cases not accurate. It needs to be revised.
- Introduction was corrected.

Reviewer 2 Report
The work is interesting. Although the authors have worked with this type of stationary phase on other occasions, the possibility of using water as the only component of the mobile phase is a great benefit. I have doubts about the behavior of these columns in dealing with complex samples with a diversity of compounds in terms of physicochemical features or what happens when applying an elution gradient. In this sense, below I present some points of discussion that should be developed in the manuscript.
As far as I understand, the authors have prepared the stationary phases in their laboratory from the Kromasil support material. The starting reagents should be listed. Besides, a brief explanation of the preparation of the columns should be given.
Would it be interesting to provide some data on the reproducibility of retention times to assess the robustness of the results? Have replicates of the injected sample been made?
I would appreciate speculative comments in the manuscript on the potential of these columns to separate complex mixtures. If you have a sample with compounds of very different polarity, how should you proceed?
The main advantage of this type of stationary phase is the use of water as the mobile phase. However, although in simple cases separation with water is feasible, in many cases a gradient elution may be necessary. Authors may comment on the circumstances and expected results.
A discussion paragraph could be added on the pros and cons regarding conventional c18 i/o other chromatographic modes.
A table similar to those for nucleosides and aromatic hydrocarbons could be added for xanthines to complete the results.
Specific comments
Title: “Phosphdiester” should be “Phosphodiester”
The terms appearing in equation 1 must be defined immediately after.
Some numbers appear with commas for decimal places
Line 185. The first sentence is general and refers to the 3 families of analytes. It should be removed from point 3.3.1 and inserted in 3.3.
Author Response
The work is interesting. Although the authors have worked with this type of stationary phase on other occasions, the possibility of using water as the only component of the mobile phase is a great benefit. I have doubts about the behavior of these columns in dealing with complex samples with a diversity of compounds in terms of physicochemical features or what happens when applying an elution gradient. In this sense, below I present some points of discussion that should be developed in the manuscript.
- We are thanks to Reviewer for critical comments. We hope, that after correction manuscript is more valuable.
As far as I understand, the authors have prepared the stationary phases in their laboratory from the Kromasil support material. The starting reagents should be listed. Besides, a brief explanation of the preparation of the columns should be given.
- Short description and proper citation are proveded.
Would it be interesting to provide some data on the reproducibility of retention times to assess the robustness of the results? Have replicates of the injected sample been made?
- Of course analyses were repeated at least three times. Information is added to the manuscript.
I would appreciate speculative comments in the manuscript on the potential of these columns to separate complex mixtures. If you have a sample with compounds of very different polarity, how should you proceed?
- Here we would like to publish the introduction of these stationary phases. Two other work are under construction, one with the separation of beta-blockers that include ion-exchange mechanism and second about adsorption investigation.
The main advantage of this type of stationary phase is the use of water as the mobile phase. However, although in simple cases separation with water is feasible, in many cases a gradient elution may be necessary. Authors may comment on the circumstances and expected results.
- The discussion was added to the manuscript. Alternative option is temperature optimization or temperature gradient. Proper literature is also provided.
A discussion paragraph could be added on the pros and cons regarding conventional c18 i/o other chromatographic modes.
- Conventional C18 stationary phase is absolutely different material. Here, significant polartity influence strongly retention mechanism. Comparison of the material with C18 is like the comparison of apples and oranges. Possibity of application in both, RP and HILIC and operation in pure water, in our opinion, confirms that it is difficult to compare it with C18.
A table similar to those for nucleosides and aromatic hydrocarbons could be added for xanthines to complete the results.
- Such table cannot provide number of theoretical plates due to unseparated peaks. Also resolution in many points is low, what is observed in the Fig. 4, If the Reviwer agree, we would like not to add this table.
Specific comments
Title: “Phosphdiester” should be “Phosphodiester”
- It is corrected, sorry for the mistake.
The terms appearing in equation 1 must be defined immediately after.
Some numbers appear with commas for decimal places
- It is corrected.
Line 185. The first sentence is general and refers to the 3 families of analytes. It should be removed from point 3.3.1 and inserted in 3.3.
- It is corrected.
Reviewer 3 Report
This manuscript “Phosphdiester stationary phases as universal chromatographic materials for separation in RP LC, HILIC, and pure aqueous mobile phase” by MikoÅ‚aj Dembek and Szymon Bocian describes chromatographic properties of stationary phases with em-295 bedded phosphodiester groups. Phosphodiester stationary phases have already been reported such as https://doi.org/10.1002/jssc.202200389 and https://doi.org/10.1016/j.talanta.2015.04.079. Hence, from the viewpoint of material science, there is no significance. If the results from chromatographic separation are significant (which I disagree with), then I would recommend submitting a manuscript to a journal in the field of separation science.
Author Response
We agreed that is not the first manuscript with phosphodiester stationary phases. However, all of them have been published by our group. We improve the materials, and in our opinion, the ability for separation in both RP LC and HILIC, as well as in pure water, is very promising. Thus, such stationary phases constitute interesting chromatographic materials and may find application. Thus, we would like to publish our results in Materials. We count on the understanding of the Reviewer.
Additionally, I would like to inform that two other Reviwers found the potential of this work. The manuscript was corrected according to Rewiewers comments.
Round 2
Reviewer 1 Report
The author did not address one of the major comments:
- The hydrophobicity of the prepared phases was characterized. However, the adsorbed water layer on the surface of the stationary phase, which is very important to HILIC separation, was not evaluated at all. Please provide data on the adsorbed water layer.
- We are working on the adsorption evaluation of tested stationary phases. We have water adsorption data, but we would like not to present it in this study. We can provide it as a material for Reviewer, not for publication.
The data on the adsorbed water layer is very important in understanding the HILIC behaviors of these stationary phases. If the authors already have the data, it should be presented here. Without the data, the manuscript is not acceptable for publication.
Author Response
The data on the adsorbed water layer is very important in understanding the HILIC behaviors of these stationary phases. If the authors already have the data, it should be presented here. Without the data, the manuscript is not acceptable for publication.
Response:
We agreed that water adsorption is necessary for HILIC mechanism. However, it was proven that polar compounds has retention in HILIC. It is the confirmation that HILIC work.
The problem is that water excess isotherm (as well as adsorption of the solute in RP LC and HILIC) is the subject of an ongoing project with a foreign partner. A separate paper is being prepared on RP and HILIC adsorption, including measurements of water adsorption isotherms.
The foreign partner does not agree to include data on water adsorption isotherms in this manuscript. How can we find a compromise with the Reviewer?
Reviewer 2 Report
I think the authors have done a good revision.
Author Response
We thank the Rewiver for help in improving the quality of our manuscript.
Reviewer 3 Report
My opinion is the same as before. The materails are not new.
Author Response
We have improved the manuscript, and we present new data that was not published.
In the present manuscript, we also present new stationary phases ( Diol-P-Chol and Diol-P-Chol) that were not characterized before.